# Cerebral blood flow is associated with matrix metalloproteinase levels during the early symptomatic phase of concussion

Nathan W. Churchill[1,2]*, Alex P. Di Battista[3,4], Shawn G. Rhind[3,4], Doug Richards[1,3], Tom A. Schweizer[1,2,5,6], Michael G. Hutchison[1,3]

1 Keenan Research Centre of the Li Ka Shing Knowledge Institute at St. Michael's Hospital, Toronto, ON, Canada, 2 Neuroscience Research Program, St. Michael's Hospital, Toronto, ON, Canada, 3 Faculty of Kinesiology and Physical Education, University of Toronto, Toronto, ON, Canada, 4 Defence Research and Development Canada, Toronto Research Centre, Toronto, ON, Canada, 5 Faculty of Medicine (Neurosurgery), University of Toronto, Toronto, ON, Canada, 6 The Institute of Biomaterials & Biomedical Engineering (IBBME) at the University of Toronto, Toronto, ON, Canada

* nchurchill.research@gmail.com

**Data Availability Statement:** The datasets analyzed for this study can be found in the figshare

## Abstract

Concussion is associated with disrupted cerebral blood flow (CBF), although there appears to be substantial inter-individual variability in CBF response. At present, the mechanisms of variable CBF response remain incompletely understood, but one potential contributor is matrix metalloproteinase (MMP) expression. In more severe forms of acquired brain injury, MMP up-regulation contributes to CBF impairments via increased blood-brain barrier permeability. A similar relationship is hypothesized for concussion, where recently concussed individuals with higher MMP levels have lower CBF. To test this hypothesis, 35 concussed athletes were assessed longitudinally at early symptomatic injury (median: 5 days post-injury) and at medical clearance (median: 24 days post-injury), along with 71 athletic controls. For all athletes, plasma MMPs were measured and arterial spin labelling was used to measure CBF. Consistent with our hypothesis, higher concentrations of MMP-2 and MMP-3 were correlated with lower global CBF. The correlations between MMPs and global CBF were also significantly diminished for concussed athletes at medical clearance and for athletic controls. These results indicate an inverse relationship between plasma MMP levels and CBF that is specific to the symptomatic phase of concussion. Analyses of regional CBF further showed that correlations with MMP levels exhibited some spatial specificity, with greatest effects in occipital, parietal and temporal lobes. These findings provide new insights into the mechanisms of post-concussion cerebrovascular dysfunction.

## Introduction

Concussion is a form of mild traumatic brain injury (TBI) that is associated with complex disturbances in brain physiology. Cerebral blood flow (CBF) has been identified as being particularly sensitive to the effects of concussion [1]. As tight control of CBF is needed to meet time-varying physiologic and neurometabolic demands, its dysregulation can have major

repository at https://figshare.com/s/60c2119945b834a17465.

**Funding:** TAS, NWC and MGH were funded by the Canadian Institutes of Health Research (CIHR) [grant numbers RN356342 – 401065, RN294001–367456]. MGH was funded by the Canadian Institute for Military and Veterans Health Research (CIMVHR) [grant number W7714-145967]. TAS and NWC were funded by Siemens Healthineers Canada.

**Competing interests:** TAS and NWC received funding from Siemens Healthineers Canada to collect data included in this study. This does not alter our adherence to PLOS ONE policies on sharing data and materials.

consequences, including more severe post-concussive symptoms and prolonged recovery [2–4]. While more severe TBI is associated with major reductions in CBF at early injury [5], the effects of concussion are more variable, with studies variously reporting mean increases in CBF relative to controls [4], mean reductions in CBF [6] and non-significant differences [7] within the first week of injury. These findings suggest substantial inter-individual variability in CBF response after concussive injury. Despite its key role in concussion, the mechanisms underlying heterogeneous CBF disturbances remain incompletely understood.

One potential mechanism of altered CBF is through the up-regulation of matrix metallo-proteinases (MMPs). MMPs are a family of intercellular calcium-dependent zinc-endopepti-dases that play key roles in cell migration, tissue development, morphogenesis and repair [8, 9]. Under pathological conditions, such as acquired brain injury, the elevated expression of these extracellular matrix-degrading proteinases may have negative physiological effects [10]. In rodent models, MMP expression tends to be increased acutely as part of the "traumatic cas-cade" and it is associated with increased permeability of the blood-brain barrier (BBB), due to the breakdown of basal lamina proteins and the tight junctions between vascular endothelial cells [11–15]. Secondary effects of increased BBB permeability include cerebral leukocyte infil-tration, vasogenic edema [16, 17] and an enhanced neuroinflammatory response [18, 19]. Col-lectively, the processes initiated by MMP up-regulation after TBI have a detrimental effect on cerebrovascular function and may similarly contribute to CBF disturbances after a concussion. To date, however, the relationship between CBF and MMP expression has been largely unex-amined in humans and for milder forms of TBI.

The present study examined associations between global CBF and peripheral MMP values for a cohort of athletes with sport-related concussion, evaluated in the early symptomatic phase of injury and at medical clearance to return to play (RTP), along with a cohort of athletic con-trols without recent injury. For both groups, a panel of plasma MMPs were measured, and mag-netic resonance imaging (MRI) was used to assess resting CBF via arterial spin labelling (ASL). This imaging technique uses magnetically-labelled arterial blood water as an endogenous tracer that tracks the delivery of oxygenated blood to brain tissues. The primary study hypothesis was that recently concussed individuals with higher MMP values would have lower global CBF val-ues (i.e., negative inter-subject correlation between these measures). It was further hypothesized that this relationship would be specific to symptomatic injury, with diminished effects among concussed athletes at RTP and among athletic controls without recent injury. If confirmed, these results would provide evidence that the variable up-regulation of MMPs is a potential mechanism contributing to heterogeneous post-concussion CBF response.

## Material and methods

### Study participants

Thirty-five (35) athletes were recruited consecutively from university-level sports teams at a single institution through the sports medicine clinic, following a diagnosis of sport-related concussion. Diagnosis was determined by staff physician following sustained direct or indirect contact to the head, with assessment of clinical features as per Concussion in Sport Group guidelines [20] and neurologic assessment including examination of cranial nerves, gait, bal-ance and gross motor function. Imaging was conducted within the first week of injury (median and interquartile range (IQR): 5 [2, 7] days) and at RTP. There was some data loss, as imaging and blood could not always be acquired concurrently. The number of participants with both imaging and blood was 27/35 at symptomatic injury and 27/35 at RTP, with 19/35 participants having data in both sessions (54% overlap). Seventy-one (71) athletes without recent concus-sion were also recruited consecutively at the start of their competitive season.

All athletes participating in the varsity program had completed baseline clinical assessments with the Sport Concussion Assessment Tool (SCAT, version 3 or 5) [21, 22] before the beginning of their season and all concussed athletes had follow-up SCAT assessments at early symptomatic injury and at RTP. None of the athletes had a history of neurological or psychiatric diseases or sensory/motor impairments. Recruitment and data collection were carried out between June 2015 and October 2017 and the study was conducted in accordance with the Canadian Tri-Council Policy Statement 2 and approval of University of Toronto and St. Michael's Hospital research ethics boards, with participants giving free and written informed consent. The datasets analyzed for this study can be found in the *figshare* repository at https://figshare.com/s/d879d6bfb929808835ad.

## Matrix metalloproteinase data

For both groups, plasma MMPs were measured, including MMP-1 (collagenase), MMP-2 and MMP-9 (gelatinases), MMP-3 and MMP-10 (stromelysins). This panel includes MMPs primarily localized to the endothelium and identified as relevant to vascular injury and TBI pathogenesis [23, 24]. Blood was drawn proximal to neuroimaging for concussed athletes (1 [0, 3] days from imaging) and controls (1 [1, 7] days from imaging) by standard venipuncture into a 10-mL $K_2EDTA$ tube, equilibrated at room temperature for one hour, and centrifuged for two minutes using a PlasmaPrep 12$^{TM}$ centrifuge (Separation Technology Inc., FL, USA). After centrifugation, plasma was aliquoted and stored at -80˚C until analysis. Blood draws were not performed on subjects who were knowingly symptomatic with a viral or bacterial infection, seasonal allergies, or on any medication other than birth control at the time of venipuncture. The MMPs were analyzed by immunoassay on the Meso Scale Discovery (MSD) sector imager 6000, using MSD 96-well MULTI-SPOT® technology (MSD®, Gaithersburg, MD, USA). MMP-1, -3, and -9, were assayed using the Human MMP 3-Plex Ultra-Sensitive kit, while MMP-2 and -10 were assayed using the Human MMP 2-Plex Ultra-Sensitive kit. All assays were run according to manufacturer's instructions, with individual samples run in duplicate.

Samples were not used for analysis if they fell outside of the range of detection provided by the manufacturer, or if the coefficient of variability (CV) between duplicate samples was greater than 25%. Given these criteria, a single value was removed for MMP-9, with remaining assays providing 100% useable data. The average CV was below 5% for all markers: MMP-1 = 4.3%, MMP-2 = 3.9%, MMP-3 = 4.2%, MMP-9 = 3.6%, MMP-10 = 3.6%. To control for the presence of heavy distribution tails that vary by biomarker (skewness: MMP-1 = 1.23, MMP-2 = -2.25, MMP-3 = 2.53, MMP-9 = 1.55, MMP-10 = 5.72; kurtosis: MMP-1 = 4.03, MMP-2 = 18.08, MMP-3 = 11.67, MMP-9 = 6.40, MMP-10 = 48.96), the values for each biomarker were winsorized at the 90th percentile (2-tailed) over all subject data, concussed and control. The adjusted biomarker distributions better approximated normality (skewness: MMP-1 = 1.06, MMP-2 = 0.24, MMP-3 = 1.26, MMP-9 = 0.93, MMP-10 = 1.10; kurtosis: MMP-1 = 3.28, MMP-2 = 2.41, MMP-3 = 3.76, MMP-9 = 3.18, MMP-10 = 3.65).

## Magnetic resonance imaging data

Athletes were imaged using a 3 Tesla MRI system (Magnetom Skyra) with standard multichannel head coil. A series of structural images were acquired, including a 3D T1-weighted magnetization prepared rapid gradient echo (MPRAGE) scan to assess neuroanatomy and facilitate the alignment of ASL scans to a common template (inversion time (TI)/echo time (TE)/repetition time (TR) = 1090/3.55/2300 ms, flip angle ($\theta$) = 8$^o$, 192 sagittal slices, field of view (FOV) = 240x240 mm, 256x256 pixel matrix, 0.9 mm slice thickness, 0.9x0.9 mm inplane, bandwidth (BW) = 200 Hz/px), a fluid attenuated inversion recovery (FLAIR) scan to

assess for lesions and tissue edema (TI/TE/TR = 1800/387/5000 ms, 160 sagittal slices, FOV = 230x230 mm, 512x512 matrix, 0.9 mm slice thickness, 0.4x0.4 mm in-plane, BW = 751 Hz/px) and a susceptibility-weighted imaging (SWI) scan to assess for micro-hemorrhage (TE/TR = 20/28 ms, $\theta$ = 15°, 112 axial slices, FOV = 193x220 mm, 336x384 matrix, 1.2 mm slice thickness, 0.6x0.6 mm in-plane, BW = 120 Hz/px). The structural scans were inspected by an MRI technologist during imaging and later reviewed by a neuroradiologist, with clinical reporting if abnormalities were identified. No abnormalities (including signs of contusion, FLAIR hyper-intensities denoting tissue edema, or SWI hypo-intensities denoting micro-hemorrhage) were found among the study participants.

Brain maps of absolute resting CBF in grey matter were obtained for each participant using a 2D pulsed ASL imaging sequence (PICORE QUIPSS II; TE/TR = 12/2500 ms, TI1/TI1s/TI2 = 700/1600/1800 ms, $\theta$ = 90°, 14 oblique-axial slices with FOV = 256x256 mm, 64x64 matrix, 8.0 mm slice thickness with 2.0 mm gap, 4.0x4.0 mm in-plane, BW = 2368 Hz/px). For this sequence, 45 tag-control image pairs were obtained, where taking the difference in signal intensity between image pairs generates a perfusion-weighted image. A single calibration image $M_0$ was also acquired, in order to rescale signal intensities into absolute units of flow. To control for noise and artifact that may confound CBF estimates, the data were processed after acquisition, using Analysis of Functional Neuroimages (AFNI; afni.nimh.nih.gov) software and customized in-house algorithms. Rigid-body correction of between-scan head movements was performed using AFNI *3dvolreg* to align tag and control scans to the calibration image $M_0$. To control for signal spikes, e.g., due to head motion, physiology and scanner noise, filtering of outlier tag-control pairs was performed using the protocol of Tan et al. [25], followed by smoothing the scans with AFNI *3dmerge* to reduce spatial noise, using a 3D Gaussian kernel with 6 mm isotropic full width at half-maximum. Voxel-wise estimates of CBF were then obtained by taking the mean of the signal difference between all tag-control pairs, rescaled into flow units of mL/100g/min based on $M_0$ values and established kinetic modelling parameters [7].

To compare regional CBF values between study participants, the brain maps were afterwards transformed into a common anatomical space, using the Montreal Neurological Institute MNI152 template as a reference. This was achieved using the FMRIB Software Library (FSL; https://fsl.fmrib.ox.ac.uk) and in-house scripts. For each participant, FSL *flirt* first computed the rigid-body alignment of their mean ASL volume to their T1-weighted image, along with the affine alignment of their T1 image to the MNI152 template. The net transform of ASL data into MNI space was then calculated with *xfm_convert* and applied to the CBF maps, resampled at 3 mm isotropic resolution. The study focused on cortical grey matter, to ensure a high signal-to-noise ratio with minimal partial volume effects when analyzing associations with MMPs. This was achieved using the Brainnetome Atlas (BNA V1.0; https://atlas.brainnetome.org) by constructing a binary mask that included all voxels with parcel labels 1 through 210. The CBF values were subsequently analyzed for all voxels in this mask.

### Analysis of clinical and demographic data

Participant demographics are listed in Table 1, including age, sex and concussion history. Clinical scores are also reported for SCAT symptoms. A symptom severity score was obtained by summing across a 22-item symptom scale, with each item receiving a 7-point Likert scale rating. A total symptoms score was also obtained by counting all symptoms with non-zero ratings. For symptom scores, non-parametric Wilcoxon tests (1-tail) determined whether post-concussion values were significantly higher than controls. Paired-measures Wilcoxon tests (1-tail) also evaluated whether post-injury values were significantly higher than pre-injury. For

**Table 1. Demographic and clinical data for controls and concussed athletes.**

| | Control | Concussion | | |
|---|---|---|---|---|
| Age (mean ± SD) | 20.0 ± 1.7 yrs. | 20.5 ± 2.2 yrs. | | |
| Female | 35/71 (49%) | 18/35 (51%) | | |
| Previous concussions | 29/71 (41%) | 18/35 (51%) | | |
| Days to RTP | – | 24 [13, 60] | | |
| Sport | Volleyball (3M/2F) | Volleyball (1F) | | |
| | Hockey (8M/17F) | Hockey (5M/5F) | | |
| | Soccer (9M/6F) | Soccer (1F) | | |
| | Football (7M) | Football (3M) | | |
| | Rugby (2M/4F) | Rugby (5M/7F) | | |
| | Basketball (4F) | Basketball (1M/1F) | | |
| | Lacrosse (6M/2F) | Lacrosse (1M/1F) | | |
| | Water polo (1M) | | | |
| | – | Mountain biking (1F) | | |
| | | Baseline | SYM | RTP |
| Total Symptoms | 2 [0, 5] | 3 [1, 4] | 8 [4, 13]** | 1 [0, 2] |
| Symptom Severity | 4 [0, 8] | 3 [1, 6] | 9 [4, 28]** | 1 [0, 2] |

All athletes were assessed at pre-season baseline, and concussed athletes were further assessed at symptomatic injury (SYM) and return to play (RTP). Clinical scores of total symptoms and symptom severity are summarized by the median and interquartile range [Q1, Q3].

A '**' denotes significant increases in symptom scores relative to baseline and controls.

all sets of statistical tests, significance was determined after adjusting for multiple comparisons at a False Discovery Rate (FDR) of 0.05.

## Analysis of MMPs and CBF

To characterize the athlete cohorts, the mean MMP and global CBF values (i.e., averaged over all grey matter voxels) were calculated for controls and for concussed athletes at symptomatic injury and RTP, along with the bootstrapped 95% confidence intervals of the mean (95%CIs), based on 1,000 bootstrap samples. The mean differences between groups were also reported at each imaging session, along with 95%CIs and $p$-values, obtained from two-sample bootstrap analyses that compared the mean values of concussed and control groups. To test the main study hypothesis, Spearman correlations between MMP values and global CBF were calculated for controls and concussed athletes at symptomatic injury and at RTP, with bootstrapped 95% CIs and $p$-values, and significant associations were identified at an FDR of 0.05. For significant MMPs and sessions, the difference in Spearman correlation between concussed and control groups $\Delta_\rho = \rho_{concussed} - \rho_{control}$ was also reported, with 95%CIs and $p$-values obtained from two-sample bootstrap analyses. Supplemental analyses examined whether the MMP-CBF correlations were confounded by demographic factors, including age, sex, prior concussion history and time post-injury. This was done by computing the partial Spearman correlation, adjusted for each covariate in turn, and testing whether there were significant changes in correlation strength, based on bootstrapped $p$-values. For all of the above analyses, missing data were handled using a pairwise deletion approach.

Additional analyses measured the effects of MMPs on regional CBF values. For the set of MMPs showing significant associations with global CBF, a single composite MMP score was generated by z-scoring and summing these variables. Spearman correlations were calculated

between the composite MMP score and CBF for each grey matter voxel, with bootstrapped $p$-values. Significant brain regions were identified by thresholding at a voxel-wise p = 0.005, followed by cluster-size thresholding at an adjusted p = 0.05, using AFNI *3dFWHMx* to estimate spatial smoothness of the CBF maps and running *3dClustSim* to obtain the minimum cluster size threshold. This procedure obtained a "consensus map" of the brain regions where MMPs affected regional CBF. The relationship was further quantified by calculating mean CBF values within significant voxels, and regressing mean CBF onto significant MMPs, with reporting of the regression coefficient $b$, coefficient of determination $R^2$ and bootstrapped 95%CIs. Regression diagnostic checks were also conducted, including plotting residuals against fitted values to assess linearity and homoscedasticity, quantile-quantile plots to assess normality of residuals and Cook's distance to test for high-leverage outliers, none of which showed evidence of substantial deviations from modelling assumptions.

Supplemental analyses examined whether taking the mean CBF over all significant clusters provided a good representative summary of cluster-specific relationships between CBF and MMP levels. For each cluster of contiguous voxels $k$, the mean CBF values were taken and separately regressed against MMP values, providing cluster-specific coefficient $b_k$. The difference score $\Delta_k = b_k - b$ was then obtained, measuring deviation of the cluster-specific model from the all-clusters model. Bootstrap resampling estimated 95%CIs and p-values of these differences, with significantly different clusters identified at an FDR of 0.05. For cluster-specific models deviating from the all-clusters model, the difference statistics were then reported.

## Results

Participant demographics are reported in Table 1 for control and concussed groups. Both cohorts are comparable in age range, proportions of male and female athletes and prior history of concussion. Concussed athletes had a median of 3–4 weeks from injury to medical clearance, which included completion of a graded exercise protocol and cognitive testing, although there was substantial variability in individual recovery times. Concussed athletes had significantly elevated total symptoms and symptom severity at early symptomatic injury, relative to their own baseline and the control cohort ($p<0.001$, all tests) at an FDR of 0.05. Symptom scores at RTP were no longer significantly elevated ($p\geq0.991$, all tests), but tended to be slightly lower than baseline.

Table 2 summarizes the distributions of MMP and global CBF values for the different groups. For all parameters, the control and concussed groups had similar group means and heavily overlapped 95%CIs. Moreover, the mean differences between groups were small relative to the group means themselves, with 95%CIs that overlapped zero. None of the

**Table 2. Average matrix metalloproteinase (MMP) concentrations and global cerebral blood flow (CBF).**

| | Control | SYM | RTP | SYM—Control | RTP—Control |
|---|---|---|---|---|---|
| **MMP-1** (pg/mL x$10^4$) | 3.13 [2.72, 3.56] | 3.42 [2.65, 4.26] | 3.41 [2.66, 4.20] | 0.29 [-0.57, 1.20] | 0.28 [-0.57, 1.18] |
| **MMP-2** (pg/mL x $10^4$) | 10.65 [10.35, 10.94] | 10.38 [10.02, 10.73] | 10.60, [10.19, 11.02] | -0.27 [-0.72, 0.19] | -0.05 [-0.55, 0.45] |
| **MMP-3** (pg/mL x$10^4$) | 1.62 [1.39, 1.86] | 1.56 [1.25, 1.91] | 1.66 [1.29, 2.07] | -0.06 [-0.47, 0.38] | 0.05 [-0.40, 0.53] |
| **MMP-9** (pg/mL x$10^4$) | 18.93 [16.82, 21.23] | 18.98 [15.48, 23.07] | 19.47 [15.75, 23.85] | 0.06 [-4.12, 4.58] | 0.55 [-3.60, 5.58] |
| **MMP-10** (pg/mL x $10^4$) | 0.21 [0.19, 0.23] | 0.20 [0.17, 0.24] | 0.21 [0.18, 0.24] | -0.01 [-0.05, 0.03] | -0.01 [-0.04, 0.03] |
| **CBF** (mL/100g/min) | 33.65 [31.68, 35.67] | 33.72 [31.31, 36.57] | 30.74 [27.32, 33.84] | -0.07 [-3.20, 3.46] | -2.91 [-6.67, 1.16] |

The mean MMP and CBF values are reported for athletic controls and for concussed athletes at early symptomatic injury (SYM) and return to play (RTP), along with bootstrapped 95%CIs of the mean. The mean differences between concussed and control groups and 95%CIs are also reported for each post-concussion imaging session. None of the comparisons between concussed and control groups attained a value of $p<0.05$, uncorrected.

comparisons attained statistical significance, with all comparisons between concussed athletes and controls having $p \geq 0.257$.

Fig 1 depicts correlation analyses of the MMP values against global CBF. For controls, the correlation values were consistently near zero, with relatively wide 95%CIs, indicating inter-subject variations in MMP and CBF values are unrelated within this cohort. In contrast, symptomatic concussed athletes showed larger negative correlations, with significant effects for MMP-2 and MMP-3 at an FDR of 0.05, indicating that higher MMP levels are associated with lower global CBF in this cohort ($p = 0.020$ and $p = 0.009$, respectively). Correlations within the symptomatic concussed group were also larger than controls, with MMP-2 having correlation difference $\Delta_\rho = -0.419$ (95%CI: -0.805, -0.011; p = 0.044) and MMP-3 having $\Delta_\rho = -0.548$ (95% CI: -0.879, -0.129; p = 0.008). At RTP, the effect was no longer no longer significant, with diminished correlations and wider 95%CIs ($p = 0.256$ and $p = 0.810$, respectively). Similarly, the values did not differ substantially from controls, with MMP-2 having correlation difference $\Delta_\rho = -0.211$ (95%CI: -0.628, 0.223; p = 0.352) and MMP-3 having $\Delta_\rho = -0.078$ (95%CI: -0.529, 0.370; p = 0.710). Supplemental analyses measured the Spearman partial correlations after adjusting for age, sex, concussion history and days post-injury. None of the tested demographic variables significantly altered correlation strength between MMPs and CBF ($p \geq 0.228$ for all analyses). These results indicate that associations between MMP expression and CBF are limited to MMP-2 and MMP-3 and to the early symptomatic phase of concussion.

For MMP-2 and MMP-3, which show significant correlations with global CBF, Fig 2 depicts their correlations with regional CBF. Fig 2A displays brain regions that have significant correlations with the (MMP-2 + MMP-3) composite score, with clusters summarized in Table 3. Significant effects were seen for extensive clusters within occipital, parietal and temporal

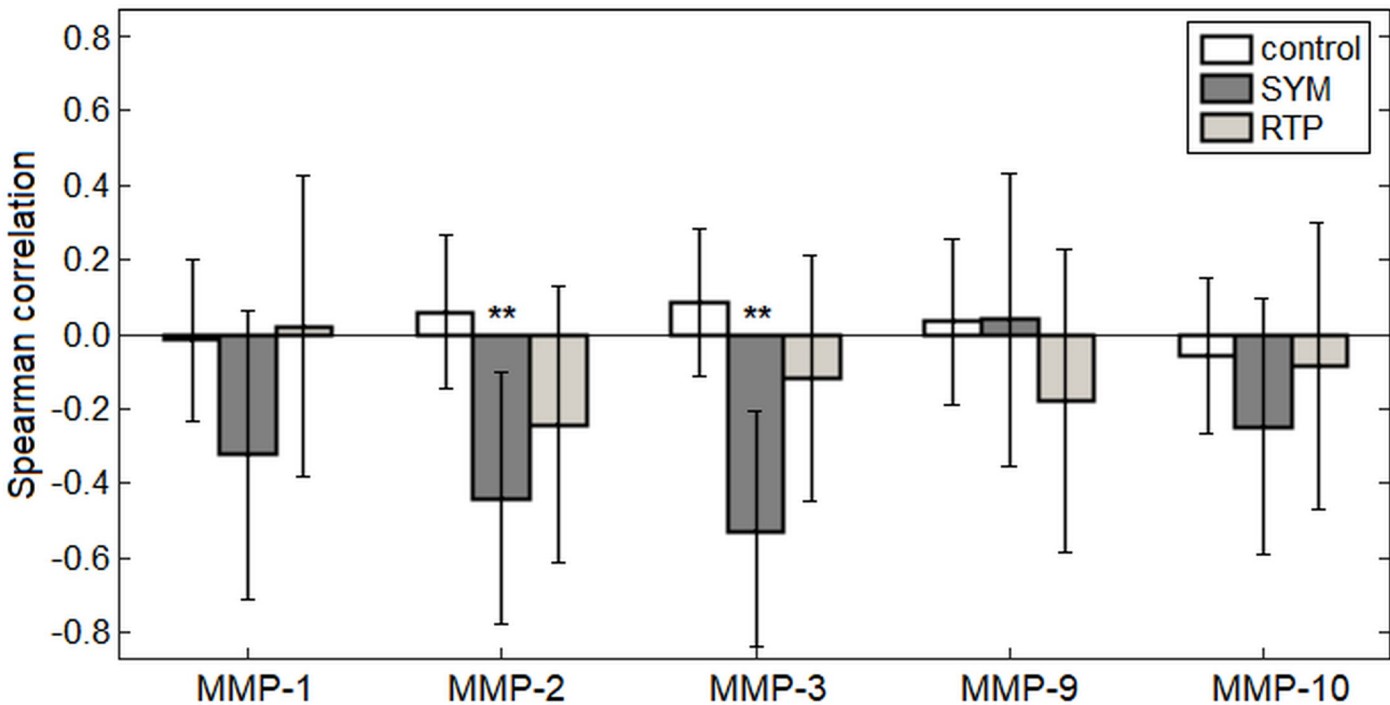

**Fig 1. Correlations between matrix metalloproteinase (MMP) concentrations and global cerebral blood flow (CBF).** Results are shown for athletic controls and for concussed athletes at early symptomatic injury (SYM) and return to play (RTP). Bars represent Spearman correlations, with error bars corresponding to bootstrapped 95%CIs. '**' denotes significant correlations (i.e., 95%CIs excluding zero) at a False Discovery Rate threshold of 0.05.

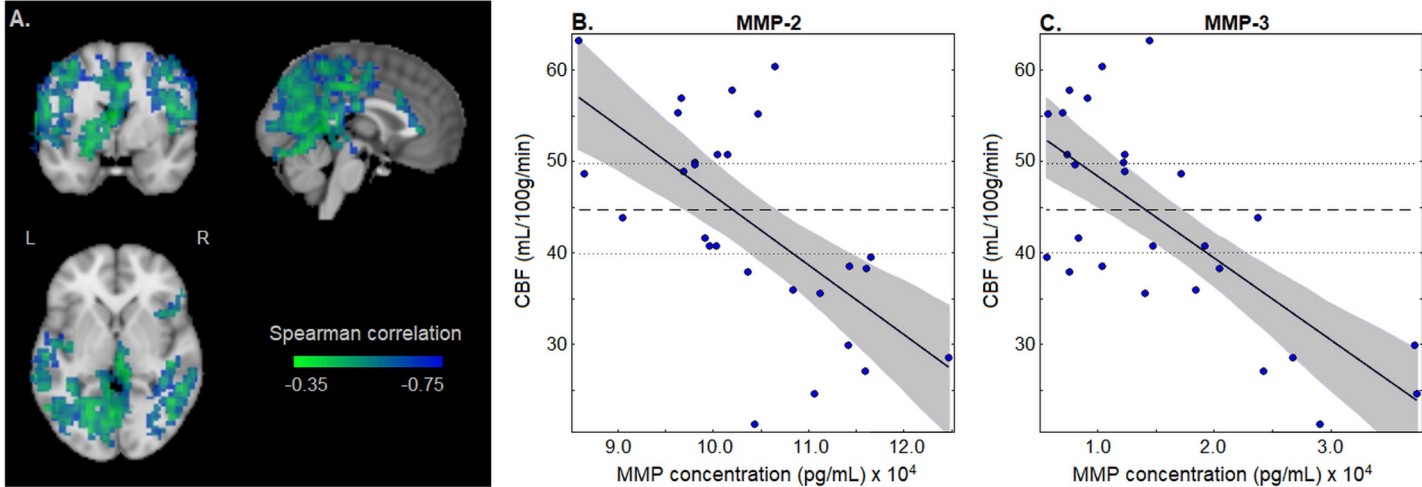

**Fig 2. Correlation between matrix metalloproteinase (MMP) concentrations and regional cerebral blood flow (CBF).** Results are shown for concussed athletes at early symptomatic injury, and for MMPs significantly correlated with global CBF (MMP-2 and MMP-3; see Fig 1). (A) Brain regions that have significant correlations with the (MMP-2 + MMP-3) composite score. Images are maximum intensity projections, centered on MNI coordinates (x = 0, y = 0, z = 0). The average CBF values of significant brain regions are also plotted against (B) MMP-2 and (C) MMP-3 concentrations. For these plots, the regression line of best fit is in solid black, with shaded bands denoting the bootstrapped 95%CIs. The thick dashed line denotes the mean CBF value for controls, with thin dashed lines enclosing the bootstrapped 95%CI of the mean.

regions. Fig 2B plots MMP-2 concentration against mean CBF, averaged over significant clusters in the brain. Linear regression obtains a coefficient $b = -7.60 \times 10^{-4}$ (95%CI: $-10.48 \times 10^{-4}$ to $-5.09 \times 10^{-4}$; p<0.001) and an $R^2$ of 0.404 (95%CI: 0.189, 0.652), and these clusters have a Spearman correlation of -0.633 (95%CI: -0.800, -0.308). Fig 2C similarly plots MMP-3 concentration against mean CBF. Linear regression obtains a coefficient $b = -8.96 \times 10^{-4}$ (95%CI: $-12.27 \times 10^{-4}$ to $-6.03 \times 10^{-4}$; p<0.001) and an $R^2$ of 0.513 (95%CI: 0.246, 0.729), and these clusters have a Spearman correlation of -0.639 (95%CI: -0.853, -0.283). Thus, associations between MMPs and CBF exhibit some spatial specificity, with greatest effects confined to posterior and dorsal brain regions. Supplemental analyses of cluster-specific CBF found no significant differences in regression coefficient values, compared to the all-cluster results. One cluster (cluster #8 in Table 2, right inferior frontal area) had a slightly higher coefficient value when regressing CBF against MMP-2 (p = 0.016) with a coefficient difference $\Delta_k = 3.65 \times 10^{-4}$ (95%CI: $0.96 \times 10^{-4}$ to $5.75 \times 10^{-4}$) but it was non-significant at an FDR of 0.05. All other clusters showed minimal

**Table 3. Cluster report for correlations between matrix metalloproteinase (MMP) concentrations and regional cerebral blood flow (CBF).** Cluster centers of mass are given in MNI coordinates and the brain regions are identified based on the nearest labelled grey matter region in the automated anatomical labelling (AAL) atlas.

| cluster | Center of mass | | | Brain region | Cluster size (mm$^3$) | Peak value (Spearman ρ) |
|---|---|---|---|---|---|---|
| 1 | -9 | -66 | 18 | Calcarine L | 24057 | -0.73 |
| 2 | 51 | -54 | 21 | Middle temporal R | 7317 | -0.68 |
| 3 | 39 | -54 | 48 | Inferior parietal R | 5697 | -0.70 |
| 4 | -60 | -33 | 18 | Superior temporal L | 4860 | -0.75 |
| 5 | -45 | -66 | 12 | Middle temporal L | 4185 | -0.67 |
| 6 | 3 | -33 | 39 | Midcingulate R | 3375 | -0.70 |
| 7 | -45 | -12 | 48 | Postcentral L | 2106 | -0.63 |
| 8 | 45 | 21 | 21 | Inferior frontal (triang. part) R | 1755 | -0.61 |
| 9 | -36 | -45 | 54 | Inferior parietal L | 1701 | -0.68 |

differences for MMP-2 (p≥0.278 for all other clusters), and there were no noteworthy deviations for MMP-3 (p≥0.096 for all clusters).

## Discussion

CBF disturbances are a key component of concussion pathophysiology, however, the underlying mechanisms are incompletely understood in humans. The present study examined whether there was an inverse relationship between peripheral MMPs and global CBF among individuals with sport-related concussion. Consistent with the main study hypothesis, MMPs were negatively correlated with global CBF for recently concussed athletes. The correlations were no longer significant at RTP or for athletic controls, further indicating that the relationship is limited to the early symptomatic phase of injury. The correlations were seen in the absence of group-level differences in mean MMP and global CBF values between concussed athletes and controls, which is consistent with our understanding of concussion as a diffuse, heterogeneous form of brain injury [7, 26]. Previous studies of sport-related concussion have often found spatially limited and study-dependent CBF effects, including increases [4], decreases [6] and non-significant differences [7] relative to controls. The variable CBF response may be due to multiple factors, including differences in time post-injury [7] and in the extent of microvascular injury [19], with the latter interpretation supported by the present MMP-related findings.

Significant negative correlations between global CBF and MMP levels are consistent with the main study hypothesis that MMP expression contributes to reduced CBF post-injury. Interestingly, the effects were not present for all measured MMPs, nor were they confined to a single subtype. MMP-2 (gelatinase) and MMP-3 (stromelysin) showed significant associations with CBF, whereas MMP-1 (collagenase), MMP-9 (gelatinase) and MMP-10 (stromelysin) did not. The identification of MMP-2 is consistent with studies of ischemic injury showing its involvement at acute injury [12, 27, 28]. Conversely, although MMP-9 has been linked with traumatic injury in rodents and is associated with greater lesion volumes and motor deficits [15, 29, 30], it did not show significant associations in this study. The mechanisms by which increased MMP expression may affect CBF are multifactorial, as MMPs are associated with injury to cerebral vasculature [15] and with BBB permeability, where the subsequent release of autocoids and free radicals can impair autoregulation [31]. In terms of secondary mechanisms, injury is followed by metabolic perturbations including ischemia, hypoxia, and vasospasm, all of which may perpetuate BBB dysfunction and edema [32, 33]. In addition, the infiltration of peripheral immune cells across the dysfunctional BBB may contribute to CBF impairments, via ongoing cellular damage and neuroinflammatory response [34, 35].

It is interesting to note diminished correlation strength between MMPs and CBF at medical clearance to RTP, which was a median of 3–4 weeks post-injury. This suggests that the underlying effects of MMPs on CBF are transient in nature, unlike some mechanisms that show persistent effects on cerebral perfusion, such as inflammatory cytokines [36, 37]. These findings are also consistent with previous studies showing dissipation of concussion-related disturbance in CBF following RTP [4, 6]. However, there are also potentially chronic effects of concussion that emerge at longer time intervals post-injury. For example, previous studies of concussion in this cohort found delayed changes in frontal CBF emerging over a year post-RTP that are potentially linked to grey matter volume loss [4, 38]. Moreover, the stroke literature suggests that elevated MMPs may confer beneficial effects chronically [39]. It may therefore be important to further examine the evolution of MMPs and CBF over longer post-concussion time intervals.

Analyses correlating the MMPs with regional CBF show broadly affected territories in the brain, as expected. The most consistently affected regions are predominantly occipital, parietal

and temporal. These are brain areas in which resting CBF also tends to be high, suggesting that they may be sensitive targets for investigating associations with MMP expression. However, it is also possible that this spatial distribution reflects brain regions experiencing the greatest MMP-mediated dysregulation of CBF. These regions are implicated in diverse processes including visuo-spatial processing and memory, which frequently show symptom impairments following concussion [22, 40]. Future research should therefore consider the potential links between MMP expression and specific post-concussion impairments.

This study fills a knowledge gap, but there are some limitations that should be acknowledged. One consideration is the post-injury time interval, as initial imaging and blood draws were conducted a median of 5 days post-injury. This enables comparisons with other imaging studies of concussion that are often on a similar timeline, but it is beyond the 24 hour window in which MMP expression is most pronounced [41, 42]. Moderately strong MMP-CBF correlations were identified, but given the rapidly evolving post-concussion neurometabolic cascade [43], these relationships may differ at acute injury. This must be verified before the present findings can be applied to emergency care settings. Another important consideration is demographic variability. Although the study found no significant effect of adjusting for age, sex or concussion history, there may be other factors that contribute to modelling error. For example, athletes with higher training loads and more frequent subconcussive blows may show greater CBF disruptions due to increased BBB permeability [44, 45], the consequences of which should be further investigated. Lastly, the detailed mechanisms linking MMP expression to CBF are not yet fully elucidated. At present, further investigation is required to directly assess the effects of proteolytic activity on cerebrovascular integrity, edema and neuroinflammatory response. The assessment of MMPs in peripheral blood also only indirectly reflects MMP levels in the CNS. Previous literature has established that cytosolic and membrane-bound proteins are released into the CSF and interstitial fluid after injury. From there, they can pass across the disrupted BBB or into the paravenous space, ultimately reaching the peripheral circulation [11, 46–48]. This provides a plausible mechanism relating peripheral MMP levels to brain injury. Nevertheless, the indirect nature of MMP measurements is a potential confound, as systemic vasculature is also a source of MMPs [49, 50], and MMP levels may be increased by, for example, peripheral nerve injury [51]. It is critical to validate the study findings, using animal models of injury or, alternatively, human studies that employ ventriculostomies to directly sample the CSF [52].

In sum, this study demonstrates that greater expression of peripheral MMP-2 and MMP-3 are correlated with reduced global CBF during the early symptomatic phase of concussive injury. These findings provide evidence for MMP expression as a potential contributor to the variable CBF response among concussed individuals. Although further research is needed to validate these findings, it is a critical first step that establishes a need for further research into the specific pathways by which MMPs and its natural inhibitors impact CBF regulation. Such research may lead to new insights into the mechanisms of CBF dysregulation and subsequent recovery, which are needed to improve clinical management and to develop targeted interventions, particularly for patients experiencing more severe or prolonged post-concussion impairments.

## Acknowledgments

We would like to thank both Ms. Maria Shiu and Ms. Katy Moes of DRDC Toronto for their technical assistance in performing the MMP assays.

## Author Contributions

**Conceptualization:** Shawn G. Rhind, Doug Richards, Michael G. Hutchison.

**Data curation:** Nathan W. Churchill, Alex P. Di Battista.

**Formal analysis:** Nathan W. Churchill.

**Funding acquisition:** Tom A. Schweizer, Michael G. Hutchison.

**Methodology:** Nathan W. Churchill, Alex P. Di Battista.

**Resources:** Shawn G. Rhind, Tom A. Schweizer, Michael G. Hutchison.

**Software:** Nathan W. Churchill, Alex P. Di Battista.

**Writing – original draft:** Nathan W. Churchill, Alex P. Di Battista.

**Writing – review & editing:** Nathan W. Churchill, Alex P. Di Battista, Shawn G. Rhind, Doug Richards, Tom A. Schweizer, Michael G. Hutchison.

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
