## [Decision Letter · Decision Letter 0]

17 Mar 2021

PONE-D-21-01481

Cerebral blood flow is associated with matrix metalloproteinase levels during the acute phase of concussion

PLOS ONE

Dear Dr. Churchill,

Thank you for submitting your manuscript to PLOS ONE. After careful consideration, we feel that it has merit but does not fully meet PLOS ONE’s publication criteria as it currently stands. Therefore, we invite you to submit a revised version of the manuscript that addresses the points raised during the review process.

As editor I support the publication of the manuscript as a full paper, however all the comments of the two reviewers need to be adequately addressed. Clear justification of the selected blood markers and stating the a priori hypothesis of the work should be given.

Since PLOS criteria are explicit on the full description of the methods in such a way that the experiments can be repeated by other groups - no need to shorten the present detailed parts.

We look forward to receiving your revised manuscript.

Kind regards,

Mária A. Deli, M.D., Ph.D.

Academic Editor

PLOS ONE

Journal Requirements:

"TAS, NWC and MGH were funded by the Canadian Institutes of Health Research (CIHR) [grant numbers RN356342 – 401065, RN294001–367456]. MGH was funded by the Canadian Institute for Military and Veterans Health Research (CIMVHR) [grant number W7714-145967]. TAS and NWC were funded by Siemens Healthineers Canada."

We note that you received funding from a commercial source: Siemens Healthineers Canada.

Reviewers' comments:

Reviewer's Responses to Questions

**Comments to the Author**

1. Is the manuscript technically sound, and do the data support the conclusions?

Reviewer #1: Partly

Reviewer #2: Yes

2. Has the statistical analysis been performed appropriately and rigorously? 

Reviewer #1: No

Reviewer #2: Yes

3. Have the authors made all data underlying the findings in their manuscript fully available?

Reviewer #1: Yes

Reviewer #2: Yes

4. Is the manuscript presented in an intelligible fashion and written in standard English?

Reviewer #1: Yes

Reviewer #2: Yes

5. Review Comments to the Author

Reviewer #1: I appreciate the time and opportunity to review this manuscript. While I found some interesting information in your manuscript, I do not think the findings are robust enough to warrant publication in this form. Since you have published analyses from this dataset previously, the analysis included in this submission might be better as a letter to the editor or a short report for the journal where the dataset was originally published. I do, however, have some comments and questions that may be helpful to you in the future.

1. In a few areas, you mention "baseline" (line 212) levels or evaluations and "pre-injury" (line 178) values. It is unclear how these are measured given the description of how you enrolled your subjects.

2. There are asterisks for Table 1, and it is unclear what these represent.

3. The use of arterial spin labeling is interesting, but I believe the description in the methods section is very technical and is laborious for the reader. Perhaps using a description that is easier to decipher may help not to lose your audience.

4. There are several "hypotheses" listed in the introduction, and I believe that one should be focused on, using it to power the study according to a primary outcome, as you have both control and treatment groups--otherwise, this is an exploratory observational study. In other words, it was unclear to me what your actual objectives were a priori.

5. line 77: I would not say that "human studies preclude. . ." as patients with severe TBI (not concussion, albeit) may have ventriculostomies and CSF sampling is possible for these biomarkers. More difficult, yes, but possible.

6. line 106: I believe the word "of" after "Imaging" could be omitted.

7. line 107-108: I find that lack of concurrent imaging and blood samples a weakness as well as the timing of the blood sampling. You mention later in the discussion that "hyperacute" sampling showed higher levels of MMP, which I think is quite clinically relevant as usually concussed patients do not stay in the hospital and would just be seen in the ED and then leave.

8. Question: Why did you choose the MMPs you chose? Also, did you consider utilizing other tight junction proteins such as occludin to confirm previously published findings. I believe that the timing of your sampling and perhaps the heterogeneity of the subjects (?) may have contributed to some findings that may lead to erroneous conclusions.

Reviewer #2: Churchill et al. investigate the correlation between concussion and peripheral blood serum MMP levels. Multiple types of MMPs were studied along with CBF imaging in the paper finding some correlation. The authors have wide experience in investigating the effects of concussion on multiple aspects of sport performance and brain functions. The paper is interesting for the field, but needs a few clarifications.

In general the acute phase of TBI is considered to be 24h after the concussion, but maximum 3-4 days in more severe cases. Although literature varies on this aspect. Therefore I would discuss the matter how the classification of acute vs. non-acute TBI is assessed more in-depth. If in this discussion it is confirmed that the used median days of the analysis here (5 days) is not considered to be "acute", I would recommend to eliminate the word "acute" from the title and include this as a limitation of the study.

I would find it necessary to show the CBF of brain regions separately and not merged, in which significant changes were found. To me it seems that in Figure 2B all regions where significant change in CBF was found, MMP and CBF values are merged to one figure. Although since part of the results and the discussion is based on the observation of spatial differences, the authors should really show the different brain regions separately, where significant change is supposed. This would increase the complexity of the paper and also would provide a more wider conclusion.

Additional comments:

- Please comment on why baseline "Symptom severity" and "Total Symptoms" are higher than RTP in Table 1.

- Reference 4 and 6 are the same, or one of them is cited wrong, please correct.

After the requested extra discussion and data analysis is performed, I will recommend the paper for publication.

6. PLOS authors have the option to publish the peer review history of their article (what does this mean?). If published, this will include your full peer review and any attached files.

Reviewer #1: No

Reviewer #2: No

---

## [Author Response · Author response to Decision Letter 0]

26 Apr 2021

Overall response: we would like to thank the reviewers for their helpful comments and suggestions. In addressing these points, we believe the clarity and scientific value of the manuscript to be significantly improved. Responses to individual reviewer points are provided below, and major changes to the manuscript text are highlighted in yellow.

REVIEWER#1

Comment 1.1: In a few areas, you mention "baseline" (line 212) levels or evaluations and "pre-injury" (line 178) values. It is unclear how these are measured given the description of how you enrolled your subjects.

Response: We apologize for the lack of clarity. All athletes participating in the varsity program had mandatory pre-season baseline clinical assessments which included administration of the SCAT. We have revised the Methods section to clarify this (p. 5, ln.107-110).

Comment 1.2: There are asterisks for Table 1, and it is unclear what these represent.

Response: This indicates post-concussion time points where symptom scores are significantly elevated relative to both controls and concussed athletes’ own baseline. We have amended the caption to state this.

Comment 1.3: The use of arterial spin labeling is interesting, but I believe the description in the methods section is very technical and is laborious for the reader. Perhaps using a description that is easier to decipher may help not to lose your audience.

Response: We have revised this section substantially to make the steps in ASL acquisition and processing clearer to a non-expert reader (p. 7-8, ln. 153-180).

Comment 1.4: There are several "hypotheses" listed in the introduction, and I believe that one should be focused on, using it to power the study according to a primary outcome, as you have both control and treatment groups--otherwise, this is an exploratory observational study. In other words, it was unclear to me what your actual objectives were a priori.

Response: Our main hypothesis is that among recently concussed individuals, those with higher MMP values will have lower global CBF values (i.e., a negative inter-subject correlation), indicating that MMP levels are related to cerebrovascular function. As a secondary hypothesis, it was predicted that the effect would be larger at early symptomatic injury, compared to both concussed athletes at RTP and uninjured controls, indicating that the effect is specific to the early phase of injury. We have revised the Introduction to focus on the main study hypothesis, as suggested (p. 4, ln. 87-91); we have also modified the Abstract (p. 2) and Discussion (p. 16, ln. 324-329; p. 17, ln. 344-345) to reinforce to the reader that this is the primary hypothesis.

Comment 1.5: line 77: I would not say that "human studies preclude. . ." as patients with severe TBI (not concussion, albeit) may have ventriculostomies and CSF sampling is possible for these biomarkers. More difficult, yes, but possible.

Response: This is an excellent point. We have amended the introduction to note that this is challenging, not impossible (p. 4, ln. 74-75). We have also added Discussion text (p. 20, ln.409-411) noting that this is a potential avenue for follow-up studies seeking to validate the present findings.

Comment 1.6: line 106: I believe the word "of" after "Imaging" could be omitted.

Response: This has been corrected.

Comment 1.7: line 107-108: I find that lack of concurrent imaging and blood samples a weakness as well as the timing of the blood sampling. You mention later in the discussion that "hyperacute" sampling showed higher levels of MMP, which I think is quite clinically relevant as usually concussed patients do not stay in the hospital and would just be seen in the ED and then leave.

Response: We agree that time of imaging and blood draws are important considerations. The present timeline is consistent with neuroimaging studies of sport-related concussion, allowing us to compare findings. As a study strength, the presence of significant correlations between MMPs and CBF indicates a relatively robust relationship that lasts beyond the acute window of injury. However, this may lead to under-estimation of effect sizes. Furthermore, generalization of these findings to the ER setting and early acute injury should be done with caution until replicated with data collected in this patient cohort. We have added these points to the discussion (p. 19, ln. 381-396). 

Comment 1.8: Why did you choose the MMPs you chose? Also, did you consider utilizing other tight junction proteins such as occludin to confirm previously published findings. I believe that the timing of your sampling and perhaps the heterogeneity of the subjects (?) may have contributed to some findings that may lead to erroneous conclusions.

Response: The panel was selected, as it includes MMPs localized mainly to the vascular endothelium and previously identified as being relevant to vascular injury and TBI pathogenesis. This has been added into the Introduction text (p. 4, ln. 84-86). Regarding the potential investigation of other tight junction proteins, we now note in the Discussion that this will be an important next step in future work to better understand the mechanistic pathways (p. 19, ln.400-401). We have also added text discussing the limitations of our imaging timeline, and potential sources of demographic heterogeneity (p. 19, ln. 381-396). Although the present study found no significant impact of adjusting for age, sex and concussion history, there are likely other unmodeled sources of heterogeneity contributing to MMP and CBF variations, such as training load and recent subconcussive blows.

REVIEWER#2

Comment 2.1: In general the acute phase of TBI is considered to be 24h after the concussion, but maximum 3-4 days in more severe cases. Although literature varies on this aspect. Therefore I would discuss the matter how the classification of acute vs. non-acute TBI is assessed more in-depth. If in this discussion it is confirmed that the used median days of the analysis here (5 days) is not considered to be "acute", I would recommend to eliminate the word "acute" from the title and include this as a limitation of the study.

Response: Since clinical and biomarker literature adheres more consistently to the 24-48 hr definition of acute injury, and our median time post-injury is 5 days, we have changed the terminology to “symptomatic injury” throughout the manuscript to avoid any potential confusion. We have also amended the discussion to note that different MMP-CBF relationships may be identified in the acute 24hr post-injury window. This warrants further investigation, particularly if the goal is to translate the present findings into emergency care settings (p. 19, ln. 381-396). 

Comment 2.2: I would find it necessary to show the CBF of brain regions separately and not merged, in which significant changes were found. To me it seems that in Figure 2B all regions where significant change in CBF was found, MMP and CBF values are merged to one figure. Although since part of the results and the discussion is based on the observation of spatial differences, the authors should really show the different brain regions separately, where significant change is supposed. This would increase the complexity of the paper and also would provide a more wider conclusion.

Response: We have conducted supplemental analyses examining whether taking the mean CBF value over all clusters yielded a good representative summary of cluster-specific associations between CBF and MMPs. For each of the contiguous clusters, we compared the cluster-specific regression coefficients to the all-clusters regression coefficient within a bootstrap resampling framework. None of the clusters deviated significantly at an FDR of 0.05, indicating that the overall mean was a good representation. One cluster did deviate at a nominal p<.05 uncorrected, and for completeness these results are reported in text. We have modified the Methods (p. 10, ln. 219-226) and Results (p. 14, ln. 291-297) to provide these details.

Comment 2.3: Please comment on why baseline "Symptom severity" and "Total Symptoms" are higher than RTP in Table 1.

Response: Although the difference is modest, this is consistent with previous publications in this cohort and prior reporting trends in the clinic. We have previously shown that this trend is correlated with improved mood state and reduced fatigue, likely due to a combination of anticipating return to play and being more physically rested at this time [Hutchison et al. (2017). JHTR, 32(3), E38-E48.] This has been noted in the Results text (p. 11, ln. 235-238).

Comment 2.4: Reference 4 and 6 are the same, or one of them is cited wrong, please correct.

Response: We have corrected the error.

---

## [Decision Letter · Decision Letter 1]

19 May 2021

PONE-D-21-01481R1

Cerebral blood flow is associated with matrix metalloproteinase levels during the early symptomatic phase of concussion

PLOS ONE

Dear Dr. Churchill,

Thank you for submitting your manuscript to PLOS ONE. After careful consideration, we feel that it has merit but does not fully meet PLOS ONE’s publication criteria as it currently stands. Therefore, we invite you to submit a revised version of the manuscript that addresses the points raised during the review process.

The manuscript has been greatly amended, there are two question related to the  scientific part which should be answered. All the other changes suggested by the reviewer are related to the structure and style of the manuscript that still need to be improved.

We look forward to receiving your revised manuscript.

Kind regards,

Mária A. Deli, M.D., Ph.D.

Academic Editor

PLOS ONE

Journal Requirements:

Reviewers' comments:

Reviewer's Responses to Questions

**Comments to the Author**

1. If the authors have adequately addressed your comments raised in a previous round of review and you feel that this manuscript is now acceptable for publication, you may indicate that here to bypass the “Comments to the Author” section, enter your conflict of interest statement in the “Confidential to Editor” section, and submit your "Accept" recommendation.

Reviewer #1: (No Response)

Reviewer #2: All comments have been addressed

2. Is the manuscript technically sound, and do the data support the conclusions?

Reviewer #1: Yes

Reviewer #2: Yes

3. Has the statistical analysis been performed appropriately and rigorously? 

Reviewer #1: I Don't Know

Reviewer #2: Yes

4. Have the authors made all data underlying the findings in their manuscript fully available?

Reviewer #1: Yes

Reviewer #2: Yes

5. Is the manuscript presented in an intelligible fashion and written in standard English?

Reviewer #1: Yes

Reviewer #2: Yes

6. Review Comments to the Author

Reviewer #1: I appreciate your attention to addressing the original comments in this manuscript. I still feel there are some style points and improvements that can be made to help this manuscript with its legibility.

I think the most significant revision in this manuscript needed is with regard to wordsmithing and I only have two questions with regard to the science.

Science questions:

1. I just don't love Figure 2B and Figure 2C. Visually there does not appear to be good correlation. I also wonder if you think there is a possibility of ASL not being a good test to use in this setting? Is there a possibly limitation to the method by which you calculated CBF and could that be a reason there is not great correlation?

2. It was unclear to me how the two MRIs had by the subjects were used together to get the data. Did all subjects have two MRIs?

Most of these are style points which I believe will make this paper more clear and of a higher quality for potential readers.

-Paragraph 3 may not need to be in the introduction; this information may need to be only in the discussion or be omitted

-Lines 83-87 likely belong in the methods.

-I know you improved the jargon with regard to ASL, but I believe it is only mentioned in the introduction now in the lines listed above, and not really explained prior to being mentioned.

-Having said that, the entirety of the methods if prone to technical jargon, and I understand that in some ways this cannot be avoided, but if possible, if you can simplify ANY of it, it will be a better read manuscript.

-Line 232: I recommend to remove "As anticipated" as this expresses conjecture, which usually is reserved for the discussion portion of the paper

-Line 234: similarly, I recommend removing "In contrast" simply because it is unnecessary.

-Lines 236-238 likely belong in the discussion as to "why?".

-Lines 249-250: there are points that belong in the discussion here

-Lines 287, 289: remove the word "moderate." It provides interpretation of results which belong in the discussion.

-Finally, in general, the discussion is very long and wordy. I would try to limit it to 5 paragraphs (currently 8), and leave out anything not pertinent to discussing your findings, with the last two paragraphs reserved for limitations and conclusions.

Reviewer #2: Authors have addressed all my comments and concerns. They modified the title according to my recommendation, that 5-days post-concussion should not be addressed as an "acute injury". I think this significantly helps readers to orientate which period after TBI was investigated. Methods were described more in detail to provide a more clear understanding of the data.

7. PLOS authors have the option to publish the peer review history of their article (what does this mean?). If published, this will include your full peer review and any attached files.

Reviewer #1: No

Reviewer #2: No

---

## [Author Response · Author response to Decision Letter 1]

22 May 2021

Overall response: we would again like to thank the reviewers for their detailed and helpful commentary, which we believe to have significantly improved the quality of the submitted manuscript. Specific responses to the remaining reviewer comments are provided below, and major changes in the manuscript text are highlighted in yellow.

REVIEWER #1

Comment 1: I just don't love Figure 2B and Figure 2C. Visually there does not appear to be good correlation. I also wonder if you think there is a possibility of ASL not being a good test to use in this setting? Is there a possibly limitation to the method by which you calculated CBF and could that be a reason there is not great correlation?

Response: Based on detailed investigation, the relationships plotted in Fig. 2B-C appear to be statistically valid. Rank-based spearman correlation with bootstrapped CIs found stable, moderately high correlations of approximately -0.63 for both MMP-2 and MMP3; this information has been added to the Results text (p. 14). Although there has been limited examination of associations between neuroimaging and serum biomarkers in concussion to date, these correlations are comparable in strength to the few extant ones (e.g., Kawata et al. (2020). Frontiers in neurology, 11; Marchi et al. (2013). PloS one, 8(3), e56805). In addition, regression diagnostics find no evidence of deviations from linearity or normality based on residual and QQ plots, nor evidence of high-leverage outliers based on the cook’s distance criterion (see figure R1 in the appended .pdf version of our response). We have added Methods text noting that these tests were performed (p. 10). 

Regarding the use of ASL to quantify CBF, it is a recognized protocol that has been validated against the “gold standard” of 15O-water PET. It shows good validity, with the added benefit that it uses arterial blood as an endogenous contrast agent and does not require any injections or exposing patients to ionizing radiation [see, e.g., Alsop, et al. (2015). Magnetic resonance in medicine, 73(1), 102-116.]. Although there are trade-offs with alternative CBF techniques (PET, SPECT, FNIRS) a detailed review is beyond the scope of this paper, particularly given the reviewer’s request to keep the discussion as concise as possible.

Comment 2: It was unclear to me how the two MRIs had by the subjects were used together to get the data. Did all subjects have two MRIs?

Response: We apologize for the lack of clarity. At a given imaging session, in addition to the ASL scan, all participants had a series of structural scans that served to characterize brain anatomy and rule out abnormalities that would complicate analysis or indicate more severe pathology. This included T1-weighted imaging to evaluate neuroanatomy, fluid attenuated inversion recovery (FLAIR) to assess for lesions and edema (appearing as hyperintensities) and susceptibility-weighted imaging (SWI) to check for signs of microhemorrhage (appearing as hypo-intense spots). This has been clarified in the Methods text (p. 7).

Comment 3: Paragraph 3 may not need to be in the introduction; this information may need to be only in the discussion or be omitted

Response: We have removed this paragraph from the Introduction text.

Comment 4: Lines 83-87 likely belong in the methods.

Response: We have moved this text to the appropriate Methods section (p. 5).

Comment 5: I know you improved the jargon with regard to ASL, but I believe it is only mentioned in the introduction now in the lines listed above, and not really explained prior to being mentioned.

Response: We have modified the Introduction to clarify what ASL is measuring (p. 4).

Comment 6: Having said that, the entirety of the methods if prone to technical jargon, and I understand that in some ways this cannot be avoided, but if possible, if you can simplify ANY of it, it will be a better read manuscript.

Response: We have made further minor edits to the Methods imaging section for clarity (p. 7-8), but we are somewhat limited by the need to provide sufficient detail about imaging and data processing that study findings can be replicated in future research.

Comment 7: Line 232: I recommend to remove "As anticipated" as this expresses conjecture, which usually is reserved for the discussion portion of the paper

Response: This line has been removed from the text

Comment 8: Line 234: similarly, I recommend removing "In contrast" simply because it is unnecessary.

Response: This line has also been removed from the text.

Comment 9: Lines 236-238 likely belong in the discussion as to "why?".

Response: This has been removed from the text, as it is not sufficiently important to justify expanding the discussion text.

Comment 10: Lines 249-250: there are points that belong in the discussion here

Response: This has been removed from the text, as it is not sufficiently important to justify expanding the discussion text.

Comment 11: Lines 287, 289: remove the word "moderate." It provides interpretation of results which belong in the discussion.

Response: This term has been removed from the Results text.

Comment 12: Finally, in general, the discussion is very long and wordy. I would try to limit it to 5 paragraphs (currently 8), and leave out anything not pertinent to discussing your findings, with the last two paragraphs reserved for limitations and conclusions.

Response: We have made significant efforts to reduce text in this section, by condensing and combining of the first 2 paragraphs (p. 16), and also by condensing and combining the 2 limitations paragraphs (p. 18), reducing the size of the discussion substantially.

---

## [Editor Report · Decision Letter 2]

31 May 2021

Cerebral blood flow is associated with matrix metalloproteinase levels during the early symptomatic phase of concussion

PONE-D-21-01481R2

Dear Dr. Churchill,

We’re pleased to inform you that your manuscript has been judged scientifically suitable for publication and will be formally accepted for publication once it meets all outstanding technical requirements.

Kind regards,

Mária A. Deli, M.D., Ph.D.

Academic Editor

PLOS ONE

Additional Editor Comments (optional):

Reviewer 2 was not available to re-review the manuscript the second time. Therefore I have checked the review and the  second revision of the manuscript. All the remaining comments were addressed and the manuscript changed as suggested. 
---

## [Editor Report · Acceptance letter]

7 Jun 2021

PONE-D-21-01481R2 

Cerebral blood flow is associated with matrix metalloproteinase levels during the early symptomatic phase of concussion 

Dear Dr. Churchill:

I'm pleased to inform you that your manuscript has been deemed suitable for publication in PLOS ONE. Congratulations! Your manuscript is now with our production department. 

Kind regards, 

on behalf of

Dr. Mária A. Deli 

Academic Editor

PLOS ONE